# A NARX Model-Based Condition Monitoring Method for Rotor Systems

**DOI:** 10.3390/s23156878

**Published:** 2023-08-03

**Authors:** Yi Gao, Changshuai Yu, Yun-Peng Zhu, Zhong Luo

**Affiliations:** 1School of Mechanical Engineering and Automation, Northeastern University, Shenyang 110819, China; gaoyi616@163.com (Y.G.); 1910658@stu.neu.edu.cn (C.Y.); 2School of Engineering and Material Science, Queen Mary University of London, London E1 4NS, UK

**Keywords:** NARX model, system identification, rotor system, condition monitoring, frequency analysis

## Abstract

In this study, we developed a data-driven frequency domain analysis method for rotor systems using the NARX (Nonlinear Auto-Regressive with eXternal input) model established by system vibration signals. We propose a model-based index of fault features calculated in a multi-frequency range to facilitate condition monitoring of rotor systems. Four steps are included in the proposed method. Firstly, displacement vibration signals are collected at multiple monitored rotating speeds. Secondly, the collected signals are processed as output data and the corresponding input data is generated. Then, NARX models are developed with input and output data to characterize the rotor system. Finally, the NRSF (Nonlinear Response Spectrum Function)-based nonlinear fault index is calculated and compared to the healthy condition. An experimental application to the misaligned rotor system is also demonstrated to verify its effectiveness. Our results indicate that the value of the index directly reflects the severity of the misaligned fault.

## 1. Introduction

Rotating machinery, such as engines, fans, water pumps, etc., plays an important role in aerospace, energy, transportation, petrochemical, and other industrial fields [1,2,3]. With the increasing use of rotating machinery in mechanical equipment, researchers have become more concerned about its performance requirements. They strive to improve the machines’ efficiency, reliability, and overall performance. However, the operating system of rotating machinery is often affected by various types of fault. Severe accidents can happen, resulting in economic losses. Therefore, it is necessary to monitor the status of the rotor system and detect early warning signs to avoid fatal damage. 

Dynamic behavior can reflect the inherent condition of the structure system. Dynamic response changes can often be used as the point of structural damage detection. Previous studies showed that faults often introduce nonlinear dynamic characteristics to rotor systems; for example, the friction fault [4,5,6], misalignment fault [7,8,9], and crack fault [10,11] cause a multi-harmonic system output response other than the fundamental harmonic component. For this reason, the fault diagnosis method for rotor systems is often accompanied by a second harmonic vibration response. The more serious the misalignment fault is, the larger the proportion of the second harmonic [8,9]. Volterra series-based methods, such as GFRF (Generalized Frequency Response Function) and NOFRF (Nonlinear Output Frequency Response Function) [12], have also been used for the fault diagnosis of rotor systems [4,13,14]. This fault diagnosis method differs from observing the second harmonic, which only considers the output signal. NOFRFs or GFRFs are identified using the system outputs and inputs to ensure that the fault features are derived from the damage to the system rather than input excitation [4,15]. In addition, we propose a modified harmonic balance-alternating frequency/time domain method to analyze the nonlinear dynamics of a rotor system [16].

NOFRFs have been proposed as an alternative frequency domain representation of GFRFs, which circles GFRFs’ multi-dimensionality problem. They are often difficult to measure and display in practice. Current approaches to Volterra series-based fault diagnosis of rotor systems also have some problems. NOFRFs require multiple tests with different amplitudes at the same speed [13]. Furthermore, data-driven-based NOFRFs evaluated from the NARX model, created using input and output data from the system, are only applicable to conditions where the input and output data are random signals [15]. It possesses the abundant frequency and amplitude characteristics of a NARX model, which can test the different properties of a system except for rotor systems. On the other hand, the truncation order of NOFRFs depends on their presentation ability and the extent to which NOFRFs can distinguish the structural properties of different damage conditions. In other words, although the NOFRFs up to the fourth order were considered in the present research, errors can still be caused if it is not appropriate a priori [14].

To solve the problems of a NOFRF fault diagnosis method, we developed a NARX model for rotor systems from the multi-harmonic signal generated by a speed-up process. We also used NRSF (Nonlinear Response Spectrum Function) [17], a system frequency analysis tool based on the notion that the system frequency response function consists of a first-order frequency response function and other orders combined. Then, we developed a condition monitoring method through an index calculated by NRSF to quantify the severity of the nonlinear fault. The proposed method is more efficient than observing the second harmonic method in the following experiment.

In summary, there are two innovations in this paper with respect to state of the art technology: (1) using Time Synchronous Averaging (TSA) technology to improve the quality of input and output signals; (2) proposing NFIs (Nonlinear Fault Indicators) to quantify the severity of the fault.

This paper is organized as follows: Section 2 describes the theoretical background of the NARX model and the NRSFs method. The condition monitoring method for rotor systems is proposed in Section 3. Section 4 verifies the validity and feasibility of this method through an experiment. Finally, our conclusions are summarized in Section 5.

## 2. The NARX Model and NRSFs of a Nonlinear System

### 2.1. The NARX Model of a Nonlinear System 

Compared to various data-driven models, NARX models are suited to condition-monitoring rotor systems due to their physically interpretable nature based on being mapped into a frequency function [18]. The NARX model represents a large class of input–output nonlinear dynamic systems [19].
(1)$$y(k)=F[y(k-1),\ldots ,y(k-n_{y}),u(k-1),\ldots ,u(k-n_{u})]+e(k)$$
where F[•] is a nonlinear function that must be identified from the input u(k) and the output y(k); k represents the discrete time; $n_{u}$ and $n_{y}$ are the maximum input and output lags; and e(k) is the model prediction error assumed to be white noise.

Considering u(k) as a polynomial function, the NARX model (1) can be written as:(2)$$y(k)=\theta_{0}+\sum_{i_{1}=1}^{n}\theta_{i_{1}}x_{i_{1}}(k)+\sum_{i_{1}=1}^{n}\sum_{i_{2}=i_{1}}^{n}\theta_{i_{1}i_{2}}x_{i_{1}}(k)x_{i_{2}}(k)+\cdots +\sum_{i_{1}=1}^{n}\cdots \sum_{i_{l}=i_{l-1}}^{n}\theta_{i_{1}\cdots i_{l}}x_{i_{1}}(k)\cdots x_{i_{l}}(k)+e(k)$$
where $\theta_{i_{1}\cdots i_{l}}$ are the model coefficients, l being the degree of polynomial nonlinearity, $n=n_{u}+n_{y}$, and
(3)$$x_{m}(k)=\{\begin{array}{ll} y(k-m) & 1\leq m\leq n_{y}\\ u(k-m+n_{y}) & n_{y}+1\leq m\leq n_{y}+n_{u} \end{array}$$

The polynomial NARX model (2) can also be represented as a linear-in-the-parameters model:(4)$$y(k)=\sum_{i=1}^{n_{M}}\theta_{i}p_{i}(k)+e(k)$$
where $n_{M}$ is the number of NARX model terms with full structure and $n_{M}=(n+L)!/\left(n!L!\right)$; $p_{i}(k),\;i=1,\ldots ,n_{M}$ are the monomials of the NARX model. Formula (2) is formulated by $x_{1}(k),\ldots ,x_{n}(k)$; $\theta_{i},\;i=1,\ldots ,n_{M}$. $n_{M}$ of the NARX model candidate is usually a high number. Identifying the NARX model helps select significant model terms from a large set of candidates based on input/output data so that a parsimony model structure can represent system dynamics.

For example, a system with a nonlinear differential equation model:(5)$$m\ddot{y}(t)+c\dot{y}(t)+k_{1}y(t)+k_{3}y^{3}(t)=u(t)$$
where u(t) and y(t) are the input and output; t is continuous time; and m, $k_{1}$, c, and $k_{3}$ are the mass, linear stiffness, damping, and nonlinear stiffness, respectively. Formula (4) can be discretized into a difference equation through a forward–backward difference approach, where $\dot{y}(t)$ and $\ddot{y}(t)$ can be discretized as follows [20]:(6)$$\dot{y}\left(t\right)=\frac{y\left(t\right)-y\left(t-\Delta t\right)}{\Delta t}\ddot{y}\left(t\right)=\frac{y\left(t+\Delta t\right)-2y\left(t\right)+y\left(t-\Delta t\right)}{\Delta t^{2}}$$
where $\Delta t=1/f_{s}$ is the time step and $f_{s}$ is the sampling frequency; therefore, (4) can be written as:(7)$$y(k)=\frac{2m-c\Delta t-k\Delta t^{2}}{m}y(k-1)-\frac{m-c\Delta t}{m}y(k-2)+\frac{\Delta t^{2}}{m}u(k-1)-\frac{k_{1}\Delta t^{2}}{m}y^{3}(k-1)$$

Let m=1kg, c=12Ns/m, $k_{1}=150N/m$, and $k_{3}=8\times 10^{8}N/m^{3}$. Formula (6) becomes (7) by setting $f_{s}=1024\mathrm{Hz}$,
(8)$$y(k)=1.9881y(k-1)-0.9883y(k-2)+9.5367\times 10^{-7}u(k-1)-762.9395y^{3}(k-1)$$
which can be considered as an identified NARX model of the Duffing system (5).

In practice, the NARX model identification typically uses the FROLS (Forward Regression Orthogonal Least Squares) algorithm by computing the ERR (Error Reduction Ratio) of each of the candidate terms [21], which will be introduced below. 

The NARX model (4) can be written into a matrix form as:(9)Y=PΘ+e
where,
(10)$$\begin{array}{c} Y={\left[y(1),\ldots ,y(N)\right]}^{T};\Theta ={\left[\theta_{1},\ldots ,\theta_{n_{M}}\right]}^{T};e={\left[e(1),\ldots ,e(N)\right]}^{T};\\ P=\left[p_{1},\ldots ,p_{n_{M}}\right]=\left[\begin{array}{ccc} p_{1}(1) & \cdots & p_{n_{M}}(1)\\ \vdots & \ddots & \vdots \\ p_{1}(N) & \cdots & p_{n_{M}}(N) \end{array}\right] \end{array}$$
with $p_{i}={\left[p_{i}(1),\ldots ,p_{i}(N)\right]}^{T},\;i=1,\ldots ,n_{M}$. A matrix letting can be orthogonally decomposed as:(11)P=WA
where A is an $N\times n_{M}$ unit upper triangular matrix $W^{T}W=\mathrm{diag}\left[d_{1},\ldots ,d_{n_{M}}\right]$, with $d_{i}=w_{i}^{T}w_{i},\;i=1,\ldots ,n_{M}$. Therefore,
(12)$$Y=(PA^{-1})(A\Theta )+e=Wg+e$$
where $W=PA^{-1}$, $g={\left[g_{1},\ldots ,g_{n_{M}}\right]}^{T}$ is an auxiliary parameter vector.

Then the ERR criterion is defined as
(13)$${\left[err\right]}_{i}=\frac{(w_{i}^{T}w_{i})g_{i}^{2}}{Y^{T}Y},\;i=1,\ldots ,n_{M}$$

The significant model term with the highest ERR value is selected by comparing the ERR values of each column vector of the regression matrix P. The process is repeated to determine the ${\bar{n}}_{M}$ significant model terms until
(14)$$1-\sum_{i=1}^{{\bar{n}}_{M}}{\left[err\right]}_{i}<\rho$$
where ρ is the threshold.

In addition, an identified NARX model can easily transform into the frequency response function, revealing core invariant behavior and facilitating the analysis, design, and fault diagnosis for the system [15].

### 2.2. The NRSFs Representation of Nonlinear System 

When a discrete-time NARX model is identified, the $NRSF_{1}(j\omega )$ [17], also known as the linear transfer function $H_{1}(j\omega )$ of the NARX model, can be directly evaluated through harmonic probing method [22] in the discrete-time domain.
(15)$$NRSF_{1}(j\omega )=H_{1}(j\omega )=\frac{\sum_{N_{u}=1}^{n_{u}}\theta_{N_{u}}\exp(-N_{u}j\omega \Delta t)}{1-\sum_{N_{y}=1}^{n_{y}}\theta_{N_{y}}\exp(-N_{y}j\omega \Delta t)}$$
where $N_{u}$ and $N_{y}$ represent lags in the linear model terms.

The Volterra functional series is shown in [18]
(16)$$y(t)=\sum_{n=1}^{+\infty }y_{n}(t)$$
where $y_{n}(t)$ is the *n*th order output of the system. Fourier transform Y(jω) of y(t) can be expressed as:(17)$$Y(j\omega )=\sum_{n=1}^{+\infty }Y_{n}(j\omega )$$
where ω is the frequency and $Y_{n}(j\omega )$ is the *n*th-order output frequency spectra obtained using Fourier transforms $y_{n}(t)$.

Therefore,
(18)$$Y(j\omega )=Y_{1}(j\omega )+\sum_{n=2}^{+\infty }Y_{n}(j\omega )=H_{1}(j\omega )U(j\omega )+\frac{Y(j\omega )-Y_{1}(j\omega )}{U(j\omega )}U(j\omega )$$
where,
(19)$$\{\begin{array}{l} NRSF_{1}(j\omega )=H_{1}(j\omega )\\ NRSF_{\infty }(j\omega )=\frac{Y(j\omega )-Y_{1}(j\omega )}{U(j\omega )}=\frac{Y(j\omega )}{U(j\omega )}-NRSF_{1}(j\omega ) \end{array}$$
are defined as the NRSFs of nonlinear systems and $NRSF_{\infty }\left(j\omega \right)$ represents the system of nonlinear dynamics [17]. Based on the NARX model (8), $NRSF_{1}(j\omega )$ can be evaluated as:(20)$$NRSF_{1}(j\omega )=\frac{9.5367\times 10^{-7}\exp(-j\omega /1024)}{1-1.9881\exp(-j\omega /1024)+0.9883\exp(-2j\omega /1024)}$$

Considering ω=[0,20] rad/s, Figure 1 shows the $NRSF_{1}(j\omega )$ and $NRSF_{\infty }\left(j\omega \right)$ of the NARX model under normalized amplitude excitation in the frequency range.

System damage usually introduces nonlinear dynamic characteristics in the structure, causing nonlinear behaviors in a system that originally presented linear characteristics [23,24]. For example, when rub-impact happens to rotor systems, the vibration waveform in the time domain is distorted because of the superposition of higher frequency components on the fundamental harmonic [4,5,7]. The vibration response of misalignment faults mainly consists of the fundamental harmonic and the second harmonic. The more severe the misalignment, the larger the proportion of the second harmonic [7,9,10]. The crack fault in the rotor system can also cause fundamental, second, and third harmonic components [10,11].

When an alternative system frequency domain expression differs from GFRF and NOFRF, NRSF divides the system’s frequency response into linear and nonlinear parts without considering the GFRF dimension or the truncation order of NOFRF. The calculation results can be accurate and reliable with a higher calculation efficiency. Previous studies verified the sensitivity of NRSFs to changes in system parameters and system faults [17], whose results show that NRSFs can reflect the changes in system characteristics and provide effective feature extraction for system analysis, parameter estimation, and fault diagnosis.

## 3. Condition Monitoring Method for Rotor Systems

### 3.1. Problem Statement

The condition monitoring method in this paper can be summarized into four steps: signal collecting, data processing, data-driven modeling, and fault feature extraction. The NARX model identification and NRSF method mentioned in Section 2 can be used to represent the rotor system and feature extraction of the model. Figure 2 shows the basic idea behind the proposed method. By extracting nonlinear characteristics from the NARX model established by the FROLS algorithm based on the vibration signals, which represent changes in physical characteristics caused by failure or damage in the rotor system, and comparing the fault index with the healthy condition, we can determine system faults and their severity. However, there are still some related problems that must be solved before they can be used for condition monitoring of rotor systems in practical engineering.

Due to noise, the quality of the input and output signals significantly affects the NARX model’s ability to characterize the system. Therefore, the measured signals should be processed before being used to establish a model. 

The basic idea of most fault diagnosis methods is to extract the fault features and compare them with the healthy state to judge the current system condition [15]. Therefore, a fault indicator can help conduct the above process more conveniently.

### 3.2. Solution of Existing Problems 

The random input that a NARX model often applies does not apply to rotor systems, which often operate at several rotating speeds [1,2,3]. The NARX model established using single harmonic signals can only be investigated under the same frequency [25]. Many researchers have analyzed rotor-bearing systems based on the speed-up process [21] and the frequency sweep system identification approach [26]. In this method, the NARX model of rotor systems is established using vibration signals under different rotating speeds, which reflect the frequency characteristics of the rotor system within the speed range.

If the rotating speeds of rotor systems in engineering practice are limited, then the range of monitoring speed $\left[\omega_{\min},\omega_{\max}\right]$ should be selected when the collecting is convenient. Collect the displacement vibration signal $x_{1}(t)$ under multiple monitoring speeds within $\left[\omega_{\min},\omega_{\max}\right]$ at the same observation point, where i represents the ith monitoring rotating speed. Keep the observation point, collecting time t, and sampling frequency $f_{s}$ unchanged for each monitoring speed.

Many data processing methods can address the noise in NARX modeling and enhance the NARX model’s generalization ability, including regularization [27], AIC (Akaike Information Criterion), BIC (Bayesian Information Criterion), etc. [28]. Time Synchronous Averaging (TSA) technology [29] efficiently alleviates the influence of noise and measurement error on signals since the rotor system’s vibration response signals are periodic. Before signal division, the vibration response signals $x_{1}(t)$ can be resampled with a sampling frequency higher than the original sampling frequency, making it more accurate to divide the signal into segments with more complete periods. The data processing method using TSA technology is presented as follows.

Determine the number of periods N for each segment of data used for stacking and divide the signal into [M/N] segments, where M represents the total number of periods. Choose zero points at the start and end of each segment. Determine the sum of each segment of data and calculate the average as follows:(21)$$y_{i}(t)=\frac{1}{\left[M/N\right]}\overset{\left[M/N\right]}{\underset{n=1}{\sum }}x_{i,n}(t)$$
where $y_{i}(t)$ is the processed data of the ith monitoring speed, $x_{i,n}(t)$ represents the nth segment of data for the ith monitoring speed.

Considering the duffing system (5), the output response under $u(t)=2\sin(\omega_{1}\cdot t)$, where $\omega_{1}=15\mathrm{rad}/s$, demonstrates the signal process method. To highlight improvements in the processed signal, a noise e(t) is mixed with the system output $x_{1}(t)$ by setting e(t) as a uniformly distributed random sequence in $[-4,4]\times 10^{-4}$. We collected 300s of the system output with a sampling frequency of $f_{s}=1024$ and obtained M=714 periods to conduct the TSA method. Figure 3 shows the number of periods for each segment is N=5 where an interval of two stars denotes a period. The signal is divided into [M/N]=142 segments. After summation averaging (15), Figure 4 shows the output $y_{1}(t)$ of using TSA technology.

The rotor system is affected by a centrifugal force from an unbalanced mass. However, the centrifugal mass and distance are sometimes unknown or difficult to measure. Therefore, the input $u_{i}(t)$ for the NARX model can be generated in a harmonic form with a normalized amplitude.
(22)$$u_{i}(t)=\sin(\omega_{i}t)$$

NARX models can adapt to different system parameters by changing the coefficients of model terms so that the amplitude of the input will not affect its ability to represent the rotor system. At each rotating speed, the sampling frequency, vibration frequency, and data length of the input signal are consistent with the output signal, and the starting point is also set to zero. The sampling frequency can be appropriately reduced after data processing to create NARX models more efficiently.

In this case, several data sets under different monitoring speeds must be handled simultaneously and one common NARX model is required. This can be achieved by the data splicing method, where the inputs $u_{i}(t)$ and outputs $y_{i}(t)$ under different speeds are spliced into U(t) and Y(t), respectively, in the same order. Then, the FROLS algorithm can be applied to establish NARX models.

To quantify the intensity of nonlinear faults and facilitate judgment of the rotor system condition, NFIs (Nonlinear Fault Indicator) are proposed in this method, which are defined as follows:(23)$$NFI=\frac{\int_{\omega_{\min}}^{\omega_{\max}}\left|NRSF_{\infty }(j\omega )\right|d\omega }{\int_{\omega_{\min}}^{\omega_{\max}}\left|NRSF_{\infty }(j\omega )\right|d\omega +\int_{\omega_{\min}}^{\omega_{\max}}\left|NRSF_{1}(j\omega )\right|d\omega }$$
where $\omega_{\min}$ and $\omega_{\max}$ are the minimum and maximum values of the frequency range, respectively. It is worth noting that NFI is a special case of NOFRF-based indices [30].

Under the investigation frequency $\omega_{n}$ selected in $\left[\omega_{\min},\omega_{\max}\right]$, we applied a single harmonic excitation with the normalized amplitude $\sin(\omega_{n}t)$ to the NARX model to calculate NRSF.

Compared to the fault features calculated at a single frequency, the index proposed in this method contains more information [15], resulting in a more reliable monitoring result.

### 3.3. Condition Monitoring Procedure

The condition monitoring method for rotor systems using NARX model and NRSF-based analysis can be summarized as follows:

Step 1: Collect the displacement vibration signals $x_{i}(t)$ under multiple monitoring rotating speeds $\omega_{i}$, where i represents the ith monitoring rotating speed;

Step 2: Process the collected signals using the TSA method and generate the corresponding input signal. Then, splice $u_{i}(t)$ and $y_{i}(t)$ into U(t) and Y(t), respectively;

Step 3: Establish a NARX model to characterize the rotor system based on input U(t) and output Y(t) data using the FROLS algorithm;

Step 4: Calculate the nonlinear fault index (17) based on the identified NARX model and compare the results to healthy conditions.

## 4. Experimental Study

### 4.1. The Misaligned Rotor System Experiment

We conducted this experiment to demonstrate the condition monitoring method using an experimental rotor system with various levels of misalignment (Figure 5).

By setting different offset distances between the edge of the bearing seat and the pedestal, we achieved different levels of misalignment fault. Each increase of 1 mm offset distance was set as a misalignment level, and a total of four levels were set in the experiment, namely level 1, level 2, level 3, and level 4, respectively. The eddy current displacement sensor was aligned with the horizontal direction of the rotating shaft and connected to the LMS mobile front end to collect the experimental rotor system’s response.

Using the proposed method in Section 3.3, we performed condition monitoring of the rotor system under the level 1 misalignment fault.

Step 1: Collect the displacement vibration signals.

The displacement vibration signals were collected under rotor system speeds of $\omega_{1}=1550\;r/min$, $\omega_{2}=1650\;r/min$, and $\omega_{3}=1700\;r/min$ at the same observation point, denoted as $x_{1}(t)$, $x_{2}(t)$, and $x_{3}(t)$, respectively, with a sampling frequency $f_{s1}=2048\;\mathrm{Hz}$ and collecting time of t=33 s;

Step 2: Process the collected signals using the TSA and data splicing methods.

The vibration response signal under $\omega_{3}=1700\;r/min$ at the level 1 misalignment demonstrates data processing in this method. The collected vibration response signal $x_{3}(t)$ was interpolated by a cubic spline and sampled again with the new sampling frequency $f_{s2}=20480\;\mathrm{Hz}$, which was L=10 times the original sampling frequency $f_{s1}$. The number of periods for each segment was determined to be N=20 and the whole M=895 periods were divided into [M/N]=44 segments. Each segment of data was summed together and an average was calculated. After the system outputs were processed, the inputs for NARX modeling were generated using (16).

The data splicing method was applied to handle the three data sets. Under the level 1 misalignment of the rotor system, the input and output data were sequentially spliced, and the data after splicing was resampled with the sampling frequency $f_{s3}=1024\;\mathrm{Hz}$. Figure 6 and Figure 7 show the splicing results of input U(t) and output Y(t), respectively.

Step 3: NARX modeling using the FROLS algorithm.

By using the FROLS algorithm, we initialized the maximum time delays for the system input $n_{u}=3$, output $n_{y}=3$, and the degree of polynomial nonlinearity l=3. Based on the input U(t) and output Y(t) data after splicing, the NARX model for level 1 of the misaligned rotor system is:(24)$$\begin{array}{cc} y(k) & =1.051y(k-1)-1.896\times 10^{-1}y(k-2)-2.708\times 10^{-1}y(k-3)\\ & +3.385\times 10^{-1}u(k-1)-6.248\times 10^{-1}u(k-2)+5.447\times 10^{-3}u{\left(k-3\right)}^{2}\\ & +3.146\times 10^{-2}u{\left(k-1\right)}^{2}+7.014\times 10^{-3}u{\left(k-3\right)}^{3}+3.047\times 10^{-1}u(k-3)\\ & -4.513\times 10^{-2}u{\left(k-2\right)}^{2}+1.834\times 10^{-1}u(k-2)y(k-1)+4.819y{\left(k-3\right)}^{2}\\ & -8.429\times 10^{-1}u{\left(k-3\right)}^{2}y(k-1)+31.137u(k-1)y{\left(k-3\right)}^{2}\\ & -20.021u(k-3)y(k-2)y(k-3)-5.318y{\left(k-1\right)}^{2} \end{array}$$

Figure 8 shows the MPO (Model Predicted Output) method’s predicted output from the NARX model (24) and the real output of level 1 of the misaligned rotor system. They are almost identical, which validates the NARX model (24).

Step 4: Calculate the NRSF-based nonlinear fault index.

The frequency domain characteristics of the NARX model were investigated in the range of monitoring rotating speeds [1550,1700] r/min. The investigation frequencies $\omega_{n}$ used to calculate NRSF were set at every other 0.5 rad/s within the range [1550,1700] r/min. Under the investigation frequency, a single harmonic excitation was applied to NARX models to calculate NRSF. Figure 9 shows NFI (17) calculations for the NARX model of the rotor system in the first misalignment level.

Using the same method as steps 1–4, we obtained the NFI of the rotor system in levels 2–4 of misalignment fault (Figure 8). As shown in Figure 9, the NFI values at four misalignment levels are 0.076, 0.21, 0.25, and 0.45, respectively. These values have an increasing monotonic trend with the severity of the misaligned fault, meaning that the index NFI’s value can directly reflect the severity of the misaligned fault. Therefore, the method proposed in this paper accurately identifies rotor system faults.

### 4.2. Comparison to the Traditional Method

Misalignment faults are often accompanied by second harmonic vibration responses. The more severe the misalignment fault is, the larger the proportion of the second harmonic [8,9]. Therefore, the amplitude of the second harmonic was always used to characterize misalignment faults in rotor systems. We compared methods to verify the advantages of the method proposed in this paper.

The vibration response spectrums of each misaligned rotor system level were calculated using the experimental data investigated beforehand. Figure 10 shows the amplitude of the second harmonic. In the traditional method, there is no obvious upward monotonic trend with the severity of the misaligned fault. For example, the amplitude of the second harmonic is gradually increasing with the severity of the misaligned fault in levels 1–3 but decreases at level 4.

The diagnosis method based on output signals is affected by system input and noise and cannot reflect the essential characteristics of the system. By contrast, we propose a model-based fault feature extraction method with an index calculated in a multi-frequency range, which is more reliable for condition-monitoring rotor systems.

## 5. Conclusions

In this study, we propose a NARX model-based NRSR method to facilitate condition monitoring for rotor systems. We used TSA technology to improve the quality of training data for NARX modeling. We also used an NFI indicator to quantify fault severity in rotor systems. Our specified conclusions are as follows:(1)NRSF-based methods are more suitable for situations where the prior conditions are insufficient and the fault characteristics are unknown since they do not have to consider the truncation order of NOFRF;(2)The NFI values at four misalignment levels were 0.076, 0.21, 0.25, and 0.45, which have a monotonic trend with the severity of the misaligned fault validating the proposed method’s efficiency over the second harmonic method.

## Figures and Tables

**Figure 1 sensors-23-06878-f001:**
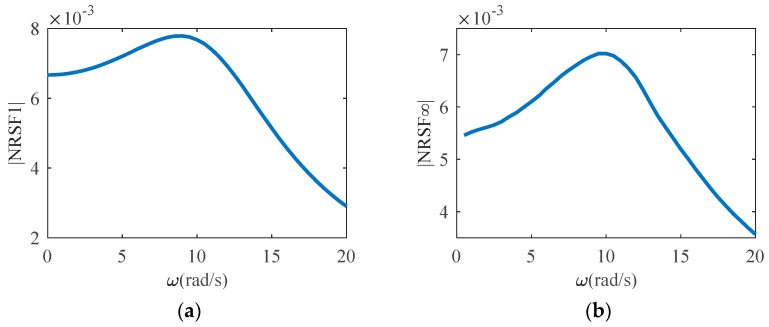
The NRSFs of the NARX model (8) (**a**) $NRSF_{1}(j\omega )$; (**b**) $NRSF_{\infty }\left(j\omega \right)$.

**Figure 2 sensors-23-06878-f002:**
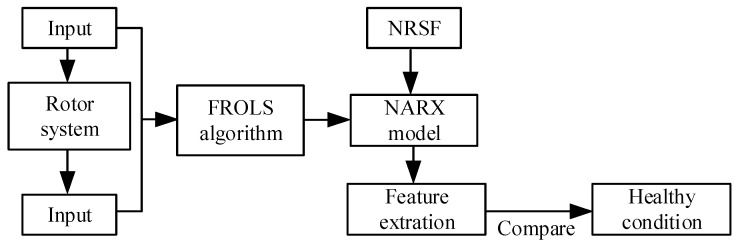
The basic idea of the proposed method.

**Figure 3 sensors-23-06878-f003:**
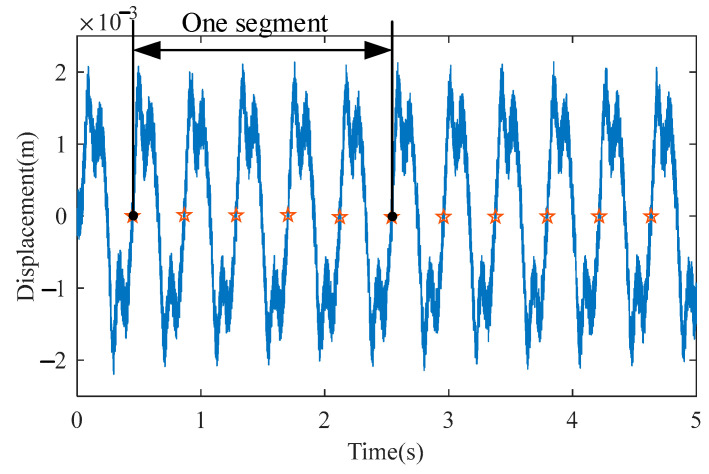
The system output $x_{1}(t)$ with noise e(t).

**Figure 4 sensors-23-06878-f004:**
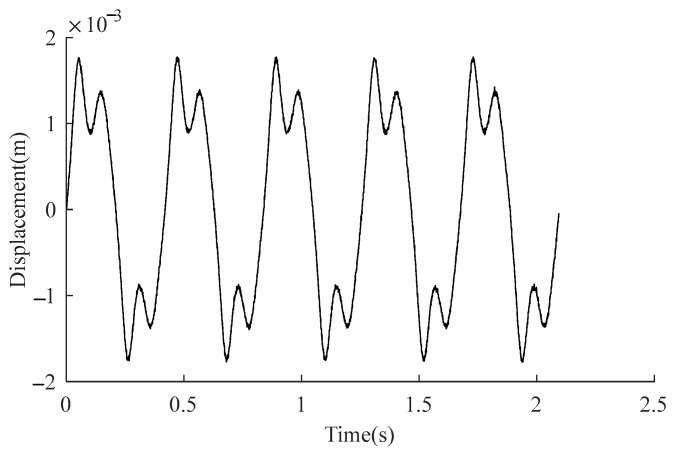
The TSA processed result of $y_{1}(t)$.

**Figure 5 sensors-23-06878-f005:**
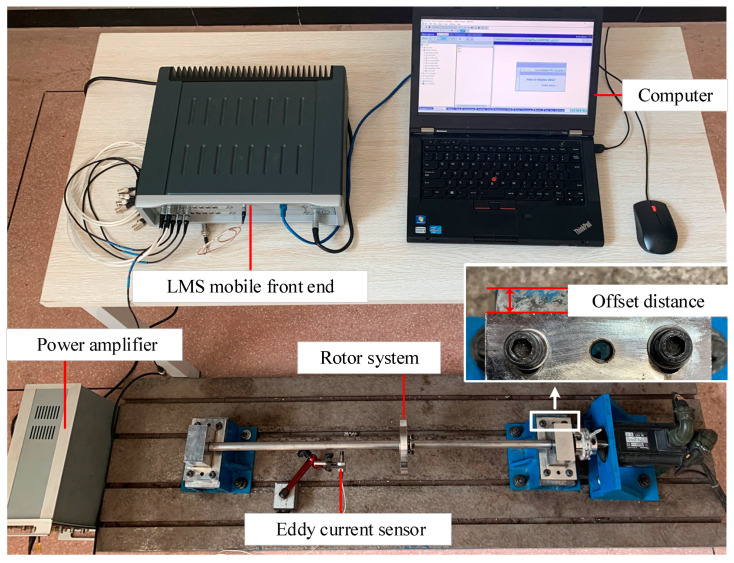
Experimental setup.

**Figure 6 sensors-23-06878-f006:**
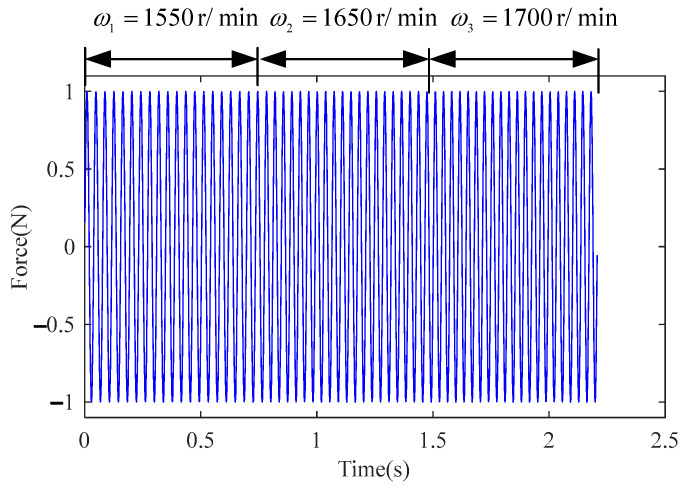
Splicing results U(t) of level 1 of the misaligned rotor system.

**Figure 7 sensors-23-06878-f007:**
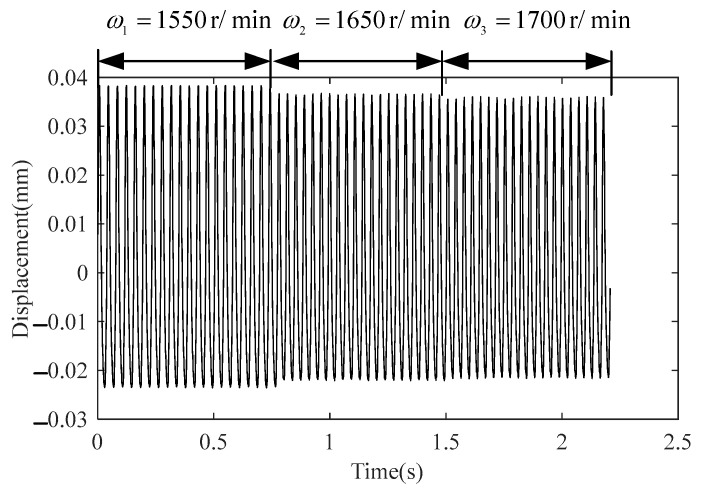
Splicing results Y(t) of level 1 of the misaligned rotor system.

**Figure 8 sensors-23-06878-f008:**
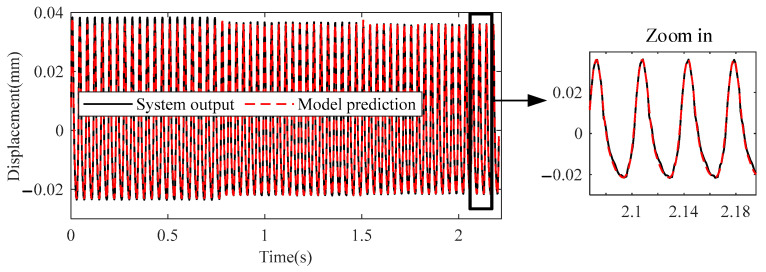
Fitting result of the first level of misaligned rotor system.

**Figure 9 sensors-23-06878-f009:**
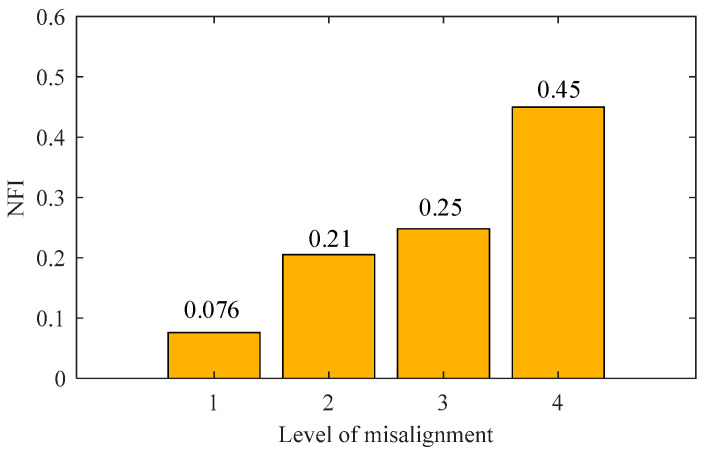
Fitting results of the first level of misaligned rotor systems.

**Figure 10 sensors-23-06878-f010:**
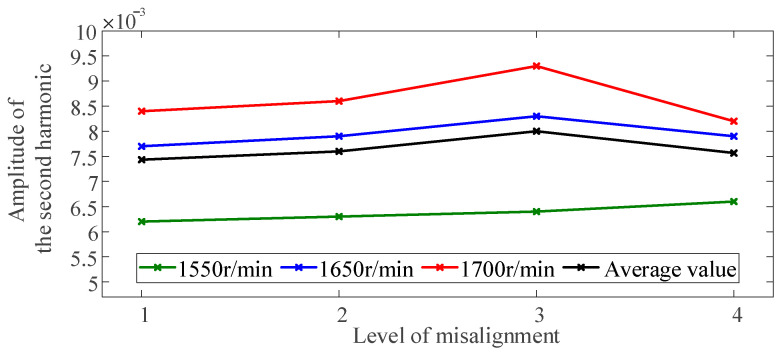
Results of the traditional method.

## Data Availability

Not applicable.

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
