# Peer review of "A NARX Model-Based Condition Monitoring Method for Rotor Systems"

_sensors, 2023, doi:10.3390/s23156878_

Round 1
Reviewer 1 Report
This paper aims to use a NARX model based NRSR method to facilitate the condition monitoring for rotor systems. First, introduce the basic idea of the condition monitoring method and clarify the existing problems of the method. The problems are: 1) the quality of the input and output signal has an essential effect on the ability of NARX model to characterize the system, 2) a fault indicator is needed to quantify the damage of the system. Then, the paper uses the TSA technology to address the first problem and proposes the NFI (Nonlinear Fault Indicator) indicator to address the second problem.
The paper is interesting. But there are some problems and suggestions.
(1) The paper uses the FROLS algorithm to identify the NARX model of the rotor system. The FROLS algorithm should provide a more detailed introduction.
(2) What is the difference of the NFI indicator proposed in this paper and traditional NOFRF-based index?
(3) The novelty points of this paper are not clear. Suggest the authors summarize the novelties with respect to the state of the art at the end of the introduction in the form of bullet points.
(4) The literature review part needs a modification by including the recently developed methods. For the alternative frequency time domain method, the followings are some examples
Nonlinear dynamics analysis of a dual-rotor-bearing-casing system based on a modified HB-AFT method. Mechanical Systems and Signal Processing, 2023, 185(15): 109805.
Reviewer 2 Report
In this manuscript, a condition monitoring method is presented for rotor systems by using NARX model system identification. The main work of this paper includes two ingredients. Use the TSA method to process the training data and propose the NFI based on NRSF to damage detection for rotor systems.
However, before the manuscript can be recommended for publication, the note below needs to be considered in a revised version.
(1) Why do the authors choose the NARX model? In fact, lots of regression models can be used instead of this model.
(2) The impact of noise is an inevitable issue in data-driven modeling. The authors need to introduce some popular and state of art algorithms for addressing the noise in NARX modelling. And why choose the TSA method in this paper?
(3) The authors should also introduce the state of art algorithms for NARX modelling of rotor systems due to the random input not applicable for rotor systems.
(4) Figure 4 is not properly displayed. The content is too small and an arrow is shown outside the figure.
Reviewer 3 Report
Overall, the authors have made a good attempt, I think. However, due to thin comparison data, the effectiveness of the proposed technique is not clear. My comments are as follows:
1. The quotation of previous articles is rough. For example, “… for example, the friction fault, misalignment fault and crack fault can cause the existence of multi-harmonic in the system output response other than the fundamental harmonic component [4-11]”. These citations are meaningless. The authors should elaborate the introduction part.
2. The research survey is not enough. The articles discussed in the introduction part are not state-of-the-art. The authors should survey past studies in detail. Besides, the authors should justify the effectiveness of the proposed method by comparing with state-of-the-art methods.
3. The authors should unify the font style. In sentences/equations, mathematical expressions should be Italic font. (Some of them are Italic fonts and others are Roman font. For example, see the imaginary number “j”.) Otherwise, readers will be confused.
4. What’s Fig. 4? The authors must check and correct Fig. 4 before submission.
5. The effectiveness of this work is not clear. Through simulations/experiments, the authors must justify the effectiveness of the proposed method by comparing with the state-of-the-art methods. Several articles are discussed in the introduction part. However, no comparison is shown with these techniques. Frankly speaking, the research survey described in the introduction part is meaningless. The authors should show comparison data.
6. The results of this research are not clear in Conclusions. The authors should show the scientific contribution of this work with concrete data.
Round 2
Reviewer 3 Report
Thank you for submitting the revised version of ID: sensors-2521931. Overall, the authors have made a good attempt. In the first version, due to thin comparison data, the reviewer failed to understand the effectiveness of the proposed technique. However, in the revised version, most of the reviewer’s requests were met by the authors. The reviewer would like to pay tribute to the authors’ great work. This is scientifically sound and contains sufficient interest to merit publication, I think.